# Implementing Indigenous Gender-Based Analysis in Research: Principles, Practices and Lessons Learned

**DOI:** 10.3390/ijerph182111572

**Published:** 2021-11-04

**Authors:** Carlos E. Sanchez-Pimienta, Jeffrey R. Masuda, Mary B. Doucette, Diana Lewis, Sarah Rotz, Hannah Tait Neufeld, Heather Castleden

**Affiliations:** 1Dalla Lana School of Public Health, University of Toronto, Toronto, ON M5T 3M7, Canada; 2School of Public Health and Social Policy, University of Victoria, Victoria, BC V8P 5C2, Canada; jeffmasuda@uvic.ca; 3Organizational Management Department, Cape Breton University, Sydney, NS B1P 6L2, Canada; marybeth_doucette@cbu.ca; 4Department of Geography, Western University, London, ON N6A 3K7, Canada; diana.lewis@uwo.ca; 5Faculty of Environmental and Urban Change, York University, Toronto, ON M3J 1P3, Canada; rotzs@yorku.ca; 6School of Public Health Sciences, University of Waterloo, Waterloo, ON N2L 3G1, Canada; hannah.neufeld@uwaterloo.ca; 7School of Public Administration, University of Victoria, Victoria, BC V8P 5C2, Canada; castleden@uvic.ca

**Keywords:** culturally relevant gender-based analysis, Indigenous Peoples, Indigenous health, renewable energy, intersectoral collaboration, decolonization, gender mainstreaming

## Abstract

Numerous tools for addressing gender inequality in governmental policies, programs, and research have emerged across the globe. Unfortunately, such tools have largely failed to account for the impacts of colonialism on Indigenous Peoples’ lives and lands. In Canada, Indigenous organizations have advanced gender-based analysis frameworks that are culturally-grounded and situate the understanding of gender identities, roles, and responsibilities within and across diverse Indigenous contexts. However, there is limited guidance on how to integrate Indigenous gender-based frameworks in the context of research. The authors of this paper are participants of a multi-site research program investigating intersectoral spaces of Indigenous-led renewable energy development within Canada. Through introspective methods, we reflected on the implementation of gender considerations into our research team’s governance and research activities. We found three critical lessons: (1) embracing Two-Eyed Seeing or *Etuaptmumk* while making space for Indigenous leadership; (2) trusting the expertise that stems from the lived experiences and relationships of researchers and team members; and (3) shifting the emphasis from ‘gender-based analysis’ to ‘gender-based relationality’ in the implementation of gender-related research considerations. Our research findings provide a novel empirical example of the day-to-day principles and practices that may arise when implementing Indigenous gender-based analysis frameworks in the context of research.

## 1. Introduction

Decades of feminist organizing at an international level have made a clear case for the need to explore and address gender inequalities in governmental policies, programs, evaluation, and research [1,2]. By the 1970s, nation-states around the globe had committed to supporting women’s empowerment and gender equality on development agendas, resulting in programs and policies that targeted women [3]. The trend in women-specific policies has shifted over the years. For instance, the 1995 Fourth United Nations’ World Conference on Women, held in Beijing, called for “mainstreaming a gender perspective in policy development and the implementation of programmes,” in part by considering the differential impacts of policy and programs on women and men [4] (p. 80). The nation-states that subscribed to the Beijing Declaration and the Platform for Action committed to developing tools to evaluate or assess progress in gender mainstreaming. Indeed, the decade of the 1990s saw the emergence of numerous tools for identifying and addressing gender inequality in governmental policies and programs [5]. To date, the name and content of the tools used by nation-states to evaluate progress on gender equality vary across the globe, with terms such as ‘Gender Impact Assessment’ being popular in the European Union and Australia, ‘Gender Analysis’ in Aotearoa/New Zealand, and ‘Gender-Based Analysis’ (herein ‘GBA’) in Canada [5,6].

Notably, the previously mentioned gender mainstreaming tools center on the concept of gender and not sex. Gender engages with the socially constructed norms, behaviours, and roles associated with gender identities (e.g., non-binary, agender, woman, man), gender expressions (e.g., feminine, masculine), and gender modalities (e.g., transgender, cisgender). In contrast, sex emphasizes the biological and physiological characteristics typically attributed to males, females, and intersex persons [7]. While sex and gender interact, gender helps in understanding people’s distinctive lived experiences through related concepts. For instance, gender identity refers to one’s sense of having a gender identity, or not, and how that identity is experienced [8]. Gender relations speak to how people interact depending on their attributed gender identities [9]. Gender diversity affirms gender identities and expressions beyond the woman/man and feminine/masculine binaries [10]. Gender expression relates to how one presents their gender identity to the world (e.g., clothing, voice modulation) [8]. Furthermore, gender modality specifies the relationship between a person’s gender identity and their sex assigned at birth [11]. The specificity of gender-related concepts can help policymakers, evaluators, and researchers act on the hierarchies and discrimination among people based on gender.

Despite the international progress on developing tools for the assessment of the implications of policy towards gender equity, the exclusion of Indigenous perspectives in gender-based policymaking is a notable deficiency across the globe [12,13]. Indigenous Peoples are the original inhabitants of territories that other ethnic groups have attempted to colonize. Indigenous nations and communities have struggled to exert their right to self-determination, their traditional lands, and for their ways of living to be respected by the nation-states who claim rights to their territories. In relation to gender mainstreaming tools, Indigenous organizations such as the Native Women’s Association of Canada (herein ‘NWAC’) have argued that GBA fails to account for the negative impacts of the historic and ongoing colonialism on Indigenous Peoples’ lives and lands [14]. Indeed, Indigenous women, Two-Spirit (a term that is used by some Indigenous persons to assert the cultural specificity of their gender identities, community responsibilities, and political organizing [15]), and gender-diverse Indigenous Peoples living within urban and rural areas of Canada continue to face racism and sexism at the socio-political, community, and interpersonal levels [16].

Grounded in an impetus to create alternatives to the Eurocentric ‘gender mainstreaming’, and on decades of scholarly and activist work of Indigenous feminist leaders and writers [17,18,19,20], a burgeoning of Indigenous approaches to Gender-Based Analysis (herein ‘Indigenous GBA’) have emerged [21,22,23]. At the heart of Indigenous GBA is a recognition of two equal and opposing forces: (1) the patriarchal histories, structures, and social norms imported from Europe that have been imposed on Indigenous communities since contact, which have had devastating consequences for their governance, community and family relations, with direct impacts on health and wellness [24,25,26]; (2) the specific cultural, geographical, historical, and spiritual contexts and strengths of diverse Indigenous communities that have survived and resisted the imposition of patriarchal worldviews [12,22]. While the above-cited Indigenous organizations and scholars have made significant headway in advancing a decolonial perspective for Indigenous GBA, from an implementation perspective there is thus far little empirical insight into the practical, everyday activities of research practitioner–learners who are attempting to implement Indigenous GBA, often from a starting point of inexperience. Such insights would be significant and timely, given the rapidly expanding and evolving state of Indigenous GBA frameworks, researchers’ highly variable levels of expertise on gender mainstreaming tools, and the contrasts between academic principles and community realities. The relevance of supporting the uptake of GBA frameworks that explicitly engage with Indigeneity is not only of relevance within Canada, but to all nation-states and research teams whose work impacts Indigenous Peoples’ lives and lands.

This paper contributes to filling this gap by sharing perspectives from the real world ‘messiness’ of critical Indigenous GBA implementation within the context of our experience in Achieving Strength, Health, and Autonomy through Renewable Energy Development for the Future (herein ‘*A SHARED Future*’); this is a multi-site research program that has prioritized gender-based considerations within a research mandate which is focused on Indigenous-centered renewable energy initiatives as a pathway to climate leadership, health equity, and reconciliation between Indigenous and Western knowledge systems. For mediating the relationships between diverse Indigenous and Western knowledge perspectives that converge within the research program, *A SHARED Future* is guided by the Mi’kmaw principle of *Etuaptmumk*, commonly known in English as ‘Two-Eyed Seeing.’ First brought forward by Elder Albert Marshall, *Etuaptmumk* refers to the gift of bringing together multiple perspectives for the benefit of all [27]. To explain *Etuaptmumk*, the Institute for Integrative Science and Health uses the visual of two connected pieces of a jigsaw puzzle, each containing an eye within them [28]. This visual illustrates the idea that Mi’kmaw knowledge and Western science are only two perspectives among the views of all cultures in the world. Sometimes one eye will have more applicable strengths than others to solve an issue, but this does not mean that one eye is better [27]. Additionally, this visual helps to explain that “no one person ever has more than one small piece of the knowledge” and that all pieces are needed [27] (p. 336). *Etuaptmumk* does not provide a series of steps to work between Indigenous knowledges and Western science; instead, their proponents offered a series of ‘lessons learned’ for weaving Indigenous knowledges and Western science together. Such lessons include doing things instead of just talking about them, weaving back and forth different ways of knowing, and working together on long-term journeys to make space for co-learning [27].

### 1.1. Toward Indigenous-Specific Gender-Based Analysis in Canada: Western Frameworks

The Government of Canada committed to implementing gender-based analysis across federal departments and agencies in 1995. The GBA framework represented one of Canada’s first efforts to centralize gender, or a gendered lens, in health projects, programs, policies, and plans. The primary concern of GBA was to explore the differential impacts of public policy towards ensuring that “the development, analysis and implementation of legislation and policies are undertaken with an appreciation of gender differences” [29] (p. 19). Unfortunately, ten years after this commitment GBA had not been systematically incorporated into policymaking within federal departments and agencies [30]. Responding to this challenge, the government of Canada progressively implemented mandatory consideration of GBA in some of its internal processes, such as the Treasury Board Submissions in 2007, the Memorandums to Cabinet in 2008, and in other federal departments from 2009 [31].

In the context of publicly funded health research, the bulk of Canada’s funding for post-secondary education institutions is administered by the Canadian Institutes for Health Research (herein ‘CIHR’), first established in 2000. Through the leadership of its Institute of Gender and Health, CIHR mandated the integration of gender- and sex-based analyses throughout the full context of its peer-review process in 2006, earlier than in other federal departments [32]. Furthermore, in 2009, the Health Portfolio of the Government of Canada—which includes CIHR—added the word ‘Sex’ at the beginning of the ‘GBA’ acronym to “emphasize the importance of sex or biological differences in the health sector” [32]. In this way, the SGBA framework sought to offer concrete methods for researchers to integrate and identify sex and gender within a variety of different contexts [33,34].

In the next decade, the implementation of gender-based frameworks continued to be fraught with multiple challenges. Notably, feminist writers criticized government officials and researchers for rarely accounting for differences among people who share the same gender identity, and overemphasizing gender to the detriment of other relevant social locations [35,36]. Responding to these issues, the GBA framework was updated; in 2011, the ‘plus’ in GBA+ was added to stress the intersectional nature of individuals’ identity factors such as race, ethnicity, religion, disability, education, sexual orientation, income, culture, and geography. The ‘plus’ also signifies how such intersecting factors influence people’s lives and the multiple forms of oppression or violence they may experience [31,37]. By taking on an intersectional approach to gender-based analysis, GBA+ attempted to identify and mitigate unintended negative impacts that stem from governmental policy and create more effective programs, policies, and services [38].

While governmental gender-based frameworks have continued to evolve in recent years, the critiques of their implementation have continued. Some have highlighted the exclusion of gender-diverse perspectives and the unintentional reinforcement of gender binaries [5,39]; others have pointed to a disconnection with contemporary feminist theory and research [40], as well as the still ongoing partial uptake of GBA+ in health research [41] and federal departments [42]. Of relevance to this paper, all stages of GBA framework development in Canada have failed to account for the distinctive understanding of gender identities, roles, and responsibilities within and across Indigenous contexts [21,26,43,44]. For instance, while GBA+ might take ‘ethnicity’ or ‘indigeneity’ as an axis of consideration, such recognition does not necessarily engage with Indigenous worldviews, nor values nation-specific conceptualizations of the relationships between people of all genders and all other beings of creation, in terms of balance, interdependence, and respect [26]. Furthermore, GBA+ lacks a theoretical grounding that guides practitioners in understanding that oppressive gender relationships are not only a consequence of patriarchy but are intimately connected to racism and colonialism as fundamental causes of injustices for people of all gender identities [26]. By not considering these key issues, using mainstream and government instituted GBA tools within Indigenous contexts runs the risk of reifying colonial understandings of gender identities and relationships, and reinforcing interventions that have been developed through Western values on Indigenous Peoples’ lives and lands [22,43].

### 1.2. Indigenous Gender-Based Analysis Frameworks

At an international level, there are significant examples of Indigenous influence on policymaking. For instance, in the Global North, Māori advisory bodies such as the Māori Women’s Welfare League and the New Zealand Māori Council have informed policymaking in Aotearoa/New Zealand [12]. In the Global South, Indigenous organizing has allied with social movement and union organizations to lead countries such as Bolivia to self-define as a pluri-national, decolonial, and anti-patriarchal nation-state, with some progress on creating policymaking that supports Indigenous women’s lives [45]. However, the dearth of Indigenous GBA frameworks at an international level is of concern; the only examples of Indigenous-specific GBA frameworks that the authors could find were those developed by Indigenous organizations in Canada.

The Canadian nation-state recognizes three groups of Indigenous Peoples–First Nations, Métis and Inuit–however, each group is internally diverse, as exemplified by the over 70 Indigenous languages currently spoken by First Nations, Métis and Inuit [46]. A total of 1.67 million people self-identified as Indigenous in the 2016 census, which accounted for 4.9% of the population [47]. Indigenous Peoples are the fastest growing and youngest sector in Canada, with 51.8% living in a metropolitan area of 30,000 people or more [47]. In response to the ongoing, and arguably glaring, exclusion of Indigenous perspectives within GBA+ policy, leading Indigenous gender-based organizations operating within Canada have worked for years to develop Indigenous-specific frameworks to serve their constituencies.

Among many examples, we highlight three notable frameworks here. The first is NWAC’s ‘Culturally-Relevant Gender-Based Analysis’ (herein ‘CRGBA’), which originated in 2007. Through a ground-breaking paper, NWAC highlighted the negative impacts of gendered discrimination against all Indigenous women and called for the urgent need to implement CRGBA in all policies, programs, and law-making [43]. Given NWAC’s mandate, CRGBA has sought to respond to the specific contexts and priorities of First Nations, Métis, and Inuit. The second is a Métis-specific, gender-based framework first developed by the Prairie Women’s Health Center of Excellence over a decade ago [48], and then further developed by *Les Femmes Michif Otipemisiwak* (Women of the Métis Nation) through the addition of an intersectionality framework with Métis-specific identity factors [21]. The third, by the Pauktuutit Inuit Women of Canada, is an Inuit GBA+ framework under development that aims to incorporate the Inuit natural laws (Maligait) and Inuit traditional knowledge (Inuit Qaujimajatuqangit) into the work of governmental departments [23]. Inuit GBA+ frameworks may consider the Inuit ways of life; the influences of the land, seasons, country food, and wildlife, contemporary influences on the Inuit ways of life; and the assessment of gender impacts in an Inuit context and under local Inuit control [49,50].

At the time of this writing, NWAC’s CRGBA appears to be the most established and widely implemented Indigenous GBA framework. CRGBA invites us to consider the historicity of gender relationships in a particular community’s context through four historical moments: before colonization, early colonization and attempted assimilation, current social and political realities, and strategies and responses looking into the future [51,52]. Furthermore, NWAC advocates for a distinction-based approach to CRGBA that pays attention to diversity within First Nations, Inuit, and Métis communities, as well as attention to the intersecting identities of Indigenous Peoples regarding their status, place of residence, and their relationships to culture, among other identity factors [22]. For the practical implementation of CRGBA, NWAC has published several resources that guide its implementation, such as a workbook [53], a pamphlet [54], and a starter kit, which outlines the four intersecting pillars of CRGBA–Indigenous knowledge, intersectionality, gender diversity, and distinction-based approaches [22]. According to NWAC, the implementation of CRGBA needs to actively incorporate Indigenous ways of knowing, as well as be trauma-informed, culturally safe, and strength-based [26].

## 2. Methodology

### 2.1. Project Context

*A SHARED Future* is a research program that brings forward stories of healing and reconciliation in the context of intersectoral partnerships associated with renewable energy transitions [55,56,57]. This research program aims to contribute to a just praxis on the global energy transition now underway in the wake of the climate change crisis. Key to our position is the recognition that not only has 400 years of colonialism led directly and inevitably to the climate change crisis, but so too does colonialism underpin many of the proposed remedies to the crisis. Within so-called Canada, Indigenous Peoples are exerting an increasingly visible leadership role in renewable energy initiatives, with one-fifth of the overall clean energy production having some degree of Indigenous involvement [58,59]. Indigenous nations and communities have diverse reasons for participating in renewable energy development, including breaking free from colonial energy systems, moving toward energy autonomy, and obtaining long-term financial benefits [56]. Mindful of historical and ongoing linkages between environmental damage on Indigenous Peoples’ lands, and gender violence toward Indigenous women, girls and gender diverse individuals that has been brought forward by the energy and ‘resource development’ sectors [50,60,61], *A SHARED Future* is working within a larger community of critical, Indigenous, and justice-oriented scholars as well as research teams to examine the intersection of gender, energy transitions, and the coming together of Indigenous and Western knowledge systems in the spirit of reconciliation.

Consisting of nine independent research projects (see http://asharedfuture.ca/our-projects/) (accessed on 13 October 2021), most of *A SHARED Future’s* independent projects are hosted by First Nations, Inuit, or urban Indigenous communities and explore diverse issues, including off-the-grid energy, food sovereignty, environmental planning, and urban sustainability. These independent research projects are located across Canada, from the west (e.g., T’Sou-ke First Nation reserves), to the east (e.g., Unama’ki), and from the south (e.g., Saukiing Anishinaabekiing) to the north (e.g., NunatuKavut). At the programmatic level, *A SHARED Future* is co-directed by two researchers and includes another four ‘Principal Investigators’ (herein ‘PIs’) and 8 ‘Community/Organizational/Governmental Co-Leads.’ As a team grant funded by the CIHR, *A SHARED Future* was mandated to integrate gender and sex into all research phases, including the appointment of a ‘Sex and Gender Champion’ within our research team. The Sex and Gender Champion was meant to possess or acquire ‘expertise in the study of sex as a biological variable and gender as a social determinant of health’ and support the research team in the implementation of sex and gender research considerations [62]—although no further direction as to how to do so was provided.

In the spirit of *Etuaptmumk*, *A SHARED Future’s* PIs established a team of Gender Champions to support the ongoing learning and implementation of Indigenous GBA across the program’s research projects. The Gender Champions participated in regular Programmatic Steering Committee and International Advisory Committee in-person meetings and video conferences, to check in with other PIs and community leaders and provide updates on the progress of their work. The Gender Champions consisted of a changing cadre of NWAC’s representatives (Sarah Harney, Project Coordinator; Hollie Sabourin, Policy Advisory; Tiffany Walsh, Senior Policy Advisor; Jaisie Walker, Senior Researcher), a non-Indigenous PI who identified as a racialized settler man and who had previously worked as an Assistant Director at CIHR’s Health and Gender Institute, and graduate students who worked with the academic PI. It is important to acknowledge that, as a partner organization and a member of the International Advisory Committee, NWAC has played an integral role within *A SHARED Future*, with the CRGBA framework being particularly influential in informing the reflections shared in this paper. The authors of this paper are researchers within the *A SHARED Future* research program. This paper draws on a series of in situ research and research support activities conducted over the first three years of the five-year research program. Our research ethics protocol was approved by Queen’s University.

### 2.2. Methods

To critically reflect on our journey of implementing culturally grounded gender considerations into *A SHARED Future’s* research projects, we drew from what we refer to as introspective methods—semi-structured internal interviews, a team sharing circle, and collaborative reflection and writing among the authors—over the past three years. Over time, these methods have involved a changing cast of academic and community team members, as well as trainees, who have come and gone since the onset of our dynamic program. Given the central role and consistent presence of six PIs in learning and facilitating the implementation of the Indigenous GBA in each *A SHARED Future* research project, their perspectives are given the most attention in the analysis that follows.

Interviews. The Gender Champions were interested in providing *A SHARED Future* PIs with an opportunity to share how they have understood and implemented gender considerations in the preparatory stages of their research. In early 2019, a PI who was part of the Gender Champions led the development of a guide for conducting semi-structured, video-call interviews. Another four PIs voluntarily agreed to participate in this research project on Indigenous GBA implementation. The interview questions explored four broad themes: previous knowledge of sex and gender issues in research, views about the Indigenous literature on sex and gender, perspectives on the interrelation between gender and colonialism, and the familiarity with Indigenous gender frameworks, including CRGBA. The interviews ranged from 44 to 66 min in duration and were transcribed verbatim. All *A SHARED Future* PIs were eligible to participate in this activity; recruitment took place from January to May in 2019.

For the data analysis, five Gender Champions did a close reading of interview transcripts guided by the questions: (1) What is the current perceived level of knowledge on culturally relevant gender considerations in research among the PIs?; (2) What are the main challenges PIs have experienced in implementing gender considerations across all phases of their research projects?; and (3) What are the PIs’ ideas for supporting each other in the practical implementation of gender research considerations for the *A SHARED Future* program and its projects? These questions guided the Gender Champions in preparing a preliminary internal report on the interview findings (Masuda et al., 2019) that was circulated within the full programmatic team, including the International Advisory Committee.

Sharing circle. Sharing circles are used as a healing and research method to gain knowledge through discussion and provide participants with opportunities for growth and transformation [63]. Common features of sharing circles include that the seating is arranged in a circle formation to emphasize that all participants join the activity as equals, a smudging ceremony is conducted at the beginning of the circle, the discussion is non-judgemental, participants hold a talking stick or feather while they speak, the object is passed around the circle either clockwise or counter clockwise depending on the specific cultural tradition of the facilitator, and there is no time limit for a participants’ contribution while they hold the talking stick or feather [63,64].

Complementing individual interview perspectives, a sharing circle was held at a program retreat in December 2019 to discuss the internal interview report and reflect on the ongoing implementation of gender considerations within *A SHARED Future’s* research projects. The sharing circle brought together the academic, as well as community leaders, of many of the independent projects to share the progress of their work and to collectively determine the directions for the research program. For about two hours, 15 academic and community project leaders spoke about their impressions of the preliminary interview report and their ideas for continuing to uphold gender considerations as a key programmatic priority for *A SHARED Future*. Instead of holding a talking stick, participants held both a voice recorder, to counteract occasional noise in the room, and a cellphone, to allow the participation of a PI who was not able to attend the circle in person.

The first round of the sharing circle data analysis involved having four Gender Champions read over the transcript. Each reader reacted to the transcript by highlighting participant quotes and commenting on their relevance to the Gender Champions’ work. The four readers had the opportunity to then view each other’s comments and their highlighted quotes. At this point, the data analysis used the same coding structure developed during the interview analysis. Based on the analytical comments of the Gender Champions, the analysis expanded to pay closer attention to the learning and implementation strategies of the *A SHARED Future* PIs and their project teams for embracing culturally relevant gender-based considerations in their independent research projects.

Collaborative, reflexive writing. Our third, and perhaps rather unorthodox, methodological approach was in the writing process itself. Collaborative writing is increasingly being used as a method of inquiry and research reporting [65,66]. In the context of this research project, the Gender Champions operationalized collaborative writing by inviting the PIs as well as a postdoctoral fellow working within *A SHARED Future* to write this research paper collaboratively from December 2020 to August 2021. Eight participants of this research project opted to have a dual role as authors and participants. This collaborative approach to authorship provided us with an opportunity to further question, refine, synthesize, and interpret our learning to date, now over halfway through the five-year mandate of our program. In addition to contributing to the intellectual facets of the writing process, each author–participant drew from their lived experience in the operationalization of Indigenous GBA principles within the *A SHARED Future* research projects they coordinate; in these instances, their voices are included *verbatim*. To support the ongoing success of the diverse *A SHARED Future* project teams, the Gender Champions agreed to center the analysis of this paper on (1) the level of preparation, resources, and strengths of the academic and community co-leads of *A SHARED Future*; and (2) the implementation strategies that *A SHARED Future* team members have employed thus far.

## 3. Findings

In this section, we offer four key insights from our ongoing endeavors to learn and implement Indigenous GBA as a multi-year, national health research program. The first two insights are grouped under an overarching theme of programmatic *research preparation*. These two themes chronicle the efforts made to operationalize the principle of *Etuaptmumk*, or Two-Eyed Seeing, among *A SHARED Future’s* participants, and refine the programmatic attempts toward building capacity on Indigenous GBA among all involved research teams. The second two insights are grouped under an overarching theme of *relational implementation*, which extends our first set of insights, speaking to the participants’ journey in grounding Indigenous GBA principles within each research teams’ local context, and to the challenges that *A SHARED Future* members faced in bringing gender-specific considerations into their work. Across all themes, we conveyed the most significant challenges, missteps, and lessons learned of the research program over its first three years, so that readers can build on our experiences to advance the promising principles and practices of Indigenous GBA. The presented participant quotes are accompanied by the initial ‘I’ if they come from interviews, ‘SC’ if they come from the sharing circle, and ‘CW’ if they come from collaborative writing; all the quotes are followed by a participant code.

### 3.1. Research Preparation: Building the Foundations for Indigenous GBA

#### 3.1.1. Bringing the Gifts of Multiple Perspectives into A SHARED Future’s Journey

As the foundation for implementing Indigenous GBA, the first theme delves into the constitution of the *A SHARED Future’s* PIs and research teams. The principle of *Etuaptmumk* guided *A SHARED Future* PIs to seek diversity across Indigenous/settler relationships to the land, gender identity, and age/professional stage. As applicants to a specific funding track focused on Indigenous ways of knowing, the initial group sought to have an Indigenous person in all leadership positions within *A SHARED Future*. At the time of grant writing, the Indigenous researchers in the team had limited availability to commit to a nominated PI role for five years. For this reason, the *A SHARED Future* PIs agreed to have a non-Indigenous researcher as the nominated PI who was willing to commit to playing a convener’s role: “*I’m a reluctant leader for sure, but you know within that first year of the program, it was clear that I needed to delegate and have shared responsibility of A SHARED Future. So, we ended up doing that*” (I-P04). *A SHARED Future* evolved into a research program that adopted a co-direction model, with one Indigenous and one non-Indigenous researcher, both women, who coordinated the work. The research program was governed through collective decision-making within a Programmatic Steering Committee formed by four Indigenous PIs and two non-Indigenous PIs. Those individual research projects that had a non-Indigenous researcher as a PI were co-led by other Indigenous researchers or community leaders. To provide guidance at a programmatic level, an International Advisory Committee was established with 11 members who identified as Indigenous or who worked for an Indigenous organization, and seven members who identified as non-Indigenous.

While *A SHARED Future* formed a diverse team across Indigeneity, gender diversity was not always achieved at all programmatic levels. For instance, a participant affirmed: “*At the beginning, we were fairly gender diverse, but right now we only have one core member of our Programmatic Steering Committee who identifies as a man and the rest identify as women*” (I-P04); additionally, no PIs identified as non-binary or agender. Reflecting on the preeminent role that people who identify as women have had in *A SHARED Future*, many participants noted that they frequently worked in research projects with most people being women: “*I’d say about 98% of participants in my studies over the years have been women, almost all women. And that hasn’t always been the objective at the outset but for several projects, it was women who came*” (SC-P01). Participants who identified as men shared some ideas into the factors that may influence men’s limited participation in these research projects, including socialization within settler–colonial contexts and their limited involvement in care work, as well as feeling uncomfortable with a lack of control: “*We’re not really great at being told what to do, and the entire principle behind community-based participatory research is research with, for, and directed by the community*” (SC-P11). Although there was less involvement of people who identify as men at some programmatic levels, this did not mean that *A SHARED Future* necessarily aimed to seek balance among gender identities across all programmatic levels. For instance, the early program feedback of the Gender Champions inspired a new research project that was not originally foreseen in the research proposal. The project was called Indigenous Women in Renewable Energy and sought to respond to an expressed need to create women-only spaces, for the sharing of their stories and discussing the gendered dynamics and implications of what has widely been acknowledged as a male-dominated renewable energy sector.

#### 3.1.2. Evolving from Written Instruction towards Relational Guidance

Given the multidisciplinary nature of the *A SHARED Future* team and the diverse research profiles of the PIs within this research program, the Gender Champions encouraged conversations on the level of familiarity of the research team members with Indigenous GBA and related bodies of academic literature. Responses from initial interviews demonstrated varying levels of Indigenous GBA awareness and experience, including familiarity with Indigenous feminist writing, experience using gender as a variable of analysis, participation in training sessions on CRGBA led by NWAC, and experience conducting gender-based analysis in the context of an environmental impact assessment. Despite participants’ diverse knowledge and experience, there was a general reluctance to claim expertise in Indigenous GBA “*I don’t want to overstate my familiarity. I’ve certainly been exposed to it, um, and I’ll leave it at that.*” (I-P04). Participants openly shared their desire to increase their knowledge on Indigenous GBA and to strengthen their capacity to implement such frameworks in the context of their research projects: “*I think that definitely providing examples of other projects in similar spheres that have done things in a good way. I think that would be really helpful to me just to give me a starting off point*” (I-P01).

The Gender Champions responded by conducting an environmental scan of the literature which focused on mainstream and Indigenous GBA that was available at the time *A SHARED Future* set out in 2017. With this information, the Gender Champions curated a ‘Living Compendium’ that extracted key principles, practices, and guiding questions for implementing Indigenous GBA [67]. The Living Compendium aimed to ensure that the PIs had the information they needed in order to center Indigenous GBA in their project proposals and was linked to an internal peer review process that supported the PIs in adopting culturally grounded gender considerations. The Compendium was a ‘living’ document in two senses: firstly, so that its principles could ‘come alive’ through their implementation in the research processes *A SHARED Future* was undertaking; second, the Compendium was intended to be updated as new resources became available. While the Living Compendium was well-received among the full project team and lauded by *A SHARED Future*’s International Advisory Committee, its practicality in facilitating the implementation of Indigenous GBA across all phases of our research projects was modest. A participant admitted: “*I would have to say, I have not read all of the information that they’ve shared with us. I’m getting more familiar; I need to do more*” (I-P03). The interviews and sharing circle showed that, by the time of data gathering, some participants had not read the Living Compendium, others had briefly reviewed it, and others intended to use it for the data analysis phases of their research projects. The Living Compendium thus failed to play its intended role as a learning resource for the PIs to implement Indigenous GBA within their research proposals. However, the participants did see value in the Living Compendium as part of a more extensive collective learning journey to promote the uptake of Indigenous GBA within *A SHARED Future*. As a team member said: “*I think that the compendium was an important first step to move toward a new place. In this way, isn’t it a good thing that we evolved beyond it?*” (CW-P07).

The internal peer review process did provide significant learning opportunities for the PIs and their research team members to center Indigenous GBA within their research proposals. Some project proposals did not incorporate gender considerations in their initial submission and were returned to their PIs with supportive feedback. For instance, a participant stated: “*When the project was conceived, they told us to consider gender and it seemed out of place. But now, at the end of the project, the issue of gender identities is emerging from the data as a point of tension for the organization*” (CW-P02). Other participants did include gender provisions within their research proposals, although their proposals tended to operationalize Indigenous GBA as part of the data analysis strategy:

“*[My research proposal] doesn’t have a specific sex and gender focus other than the fact that some of the work that some of the participants in the study will be doing is more of a gender equity focus. So, I’m really hoping to delve into that in terms of the results (…) and that I will have some support from the sex and Gender Champions that are on the team*” (I-P01).

While the participants frequently suggested that providing technical support to all the research teams in data analysis was a key role that the Gender Champions could play, the Gender Champions admitted they did not have the capacity to provide that level of support. Instead, the Gender Champions suggested that the PIs explore and discuss more relational and value-driven components of the Indigenous GBA frameworks rather than limiting themselves to the analysis considerations:

“*As soon as we say that it’s all about analysis, we fall into the trap of a Western gaze and to think ‘oh, this is just about making sure that we have enough data points of men and women in our analysis’ and that’s only a small part of it. It’s also about positionality, relationships, practice and commitments.*” (SC-P05).

### 3.2. Relational Implementation: Grounding the Principles of Indigenous GBA within Our Community Relationships

#### 3.2.1. Bringing to the Forefront our Relationships with Research Partners

To foster increased attention to the key role of relationality within Indigenous GBA, the Gender Champions encouraged participants to reflect with their community partners about the specific history, culture, language, and spirituality that grounded each research project. Such grounded expertise could become the primordial source of guidance for each research team to operationalize Indigenous GBA. In other words, it was not necessary to bring in someone who identified themself as an expert in Indigenous GBA for guiding each research team. Rather, lived experience was the foundational source of knowledge that could guide each team envisioning Indigenous GBA implementation: “*Maybe it’s just about bringing someone else into the Advisory Committee that could have more insight on this topic from lived experience and starting a conversation: Hey, have you noticed that we only have women in the room? What is happening?*” (SC-P09). Hosting informal conversations about aspects of interest or concern could be an initial step towards obtaining local direction on how to address the gendered implications of each research project. Depending on the communities and organizations each research team worked with, the participants identified relevant stakeholders who could guide the implementation of Indigenous GBA. For instance, a participant highlighted the key role that a grandmothers’ group had in their research project: “*[First Nation] has a really strong grandmothers’ group (...) That’s been the discussion from the very first meeting [for our research project], that we have to make sure that the grandmothers are involved.*” (I-P03). Similarly, other participants noted that women, Indigenous 2SLGBTQ+ mentors, and youth who were already active advocates within their communities on gender diversity could guide the local implementation of Indigenous GBA: “*Youth hold the most progressive attitudes toward gender. (…) Therefore it should be seen as a primary concern of a research project to provide both voice and protection for youth and their perspectives to emerge.*” (CW-P05).

Having recognized the rich diversity of knowledge and experience among the *A SHARED Future* research partners, participants also admitted that seeking advice from community partners was not necessarily a straightforward process. For example, a participant talked about the implications of using CRGBA within the context of the Indigenous community she was a part of: “*What does gender identity mean in culture? And how do we make sure we’re not automatically coming at this analysis from a colonial mindset because we’re in the academy, using the English Western dominant worldview?*” (I-P02). The same participant affirmed that this question was complex to answer and would require Indigenous languages, back-to-back translation and a significant time spent with Elders to figure it out. Similarly, other participants wondered what Indigenous GBA could look like when the communities they worked with had languages that do not use gendered third-person pronouns, or that were traditionally structured through matrilineal kinship systems. While the answers to these questions were not clear for participants during data collection, they identified critical areas that required community guidance and that they could work on throughout their research processes.

For the case of the participants whose work was conducted in relationship with partners in the extractive and renewable energy sectors, other types of challenges appeared, such as men’s privilege and anti-Indigenous racism. A participant who identified as an Indigenous woman said: “*I’m shocked at how even in the first two interviews that we’ve done [with a community organization], there’s this very real sense of THE white man in the room* vs. *the Indigenous folks or the women in the room*” (SC-P02). The participants admitted that not all of the research partnerships were welcoming of the relational approaches that the implementation of Indigenous GBA necessitates: “*Outside of the general social sciences and the arts and the environment where you tend to be able to talk about touchy-feely things, you are going to be met with resistance, period*” (SC-P02). Another participant, who identified as a non-Indigenous man, agreed that in his interviews with a government department, asking questions about gender could feel awkward: “*There is a significant blind spot in terms of the men involved in the project, and that’s where it gets awkward because now we are putting them on the spot and shining the light on their gender blind spots*” (SC-P06). Navigating oppressive power relationships in the context of fieldwork was a significant challenge shared by some of the participants. Indeed, there was a sense that increased resources on how to address these relational challenges in the context of Indigenous GBA implementation could be an area of further development: “*This seems to be an example of where more deliberate processes or procedures could be helpful. Would it be useful to have ‘conversation starters or strategies for bringing privileges and power into conversations in safe or respectful ways?*” (CW-P02).

#### 3.2.2. Recognizing and Addressing Gender Blind Spots within A SHARED Future

The last theme speaks to the challenges of having conversations about gender within *A SHARED Future*. From the perspective of some participants who identified as men, it was difficult to see the oppressive gender relationships that prevail in Western contexts operating within their home communities: “*At home, we never think about this because we are all family. So, if anything, when I go out on a canoe, we don’t even think about gender (…) it doesn’t matter if you are male or female*” (SC-P10). Contrastingly, several participants who identified as women emphasized that there were ongoing challenges for women and gender-diverse individuals in both the community and professional contexts they were a part of. The participants perceived a sense of tension within the sharing circle when these contrasting views emerged. Recognizing that conversations on Indigenous GBA implementation can be severely hindered when participants refuse to acknowledge their privileges, or dismiss the importance of having these conversations, a participant offered the following reflection: “*What if you knew that you were unintentionally encouraging women in your life to regularly go into a place or places where they will be disrespected? How does that make you feel, and would you still do it?*” (SC-P02). Other participants who identified as women offered further suggestions for all of the participants to engage in respectful and transformative conversations about the gendered implications of the research. Such suggestions included hosting activities that promote trust-building, being open to sharing vulnerabilities, acknowledging our privileges, attentive listening, mutual accountability, and hosting conversations from a place of love and mutual care.

Despite the tensions that emerged in the sharing circle, the participants agreed that taking the time for having conversations that centered on gender had a great value. These types of conversations were seen as necessary because they were often absent within other research teams or held exclusively within academic terms with no input from community perspectives. For *A SHARED Future*, opening spaces for hosting gender-focused conversations represented a small but significant step towards holding each other more accountable towards fostering balanced gender relationships within our research projects. While the tensions between research partners or among members of *A SHARED Future* were not resolved at the point of data gathering, a participant highlighted that the questions and matters of concern that were raised in the interviews and sharing circle were important steps forward in the team’s collective journey of implementing Indigenous GBA:

“*I don’t know that we need an answer today or I don’t know if this is something that we need to force somebody to answer. I think raising the questions, making people think, and then reflecting on it is powerful as well to (...) sit with it for a little while to really reflect and give space for those incremental changes that are 150 years in the making.*” (SC-P02).

## 4. Discussion

We have written this paper to share the collective learnings of one large, multi-site research team’s efforts to center gender within a decidedly decolonial research praxis. The context of our work is of Indigenous leadership within renewable energy development in Canada, but we believe the relevance of our experiences applies to other international research contexts that purport to confront and dismantle colonial patriarchal systems. Our introspective research process, which is still ongoing, has revealed critical insights into the implementation of Indigenous GBA in the context of environmental health research, which was gained from both successes and failures as we feel our way forward. The day-to-day activities of Indigenous GBA learning and implementation included building a team that brought forward the gifts of multiple perspectives, compiling and sharing a Living Compendium of existing Western and Indigenous GBA frameworks, providing feedback on the implementation of gender considerations within individual research proposals, seeking guidance from the Indigenous collaborators that each project worked with, and discussing the challenges for the implementation of Indigenous GBA among the *A SHARED Future* membership. To the authors’ knowledge, this paper presents the first empirically focused, reflexive case study of what Indigenous GBA may look like as a learning praxis in a research initiative of this scale. Inspired by *Etuaptmumk*, we share three major lessons that emphasize the importance of a relational and learning-based approach within Indigenous GBA.

Firstly, in designing research projects through the lens of Indigenous GBA, it is fundamental to consider whose voices are represented within the research team. Research findings showed that the principle of *Etuaptmumk* encouraged the *A SHARED Future*’s membership to consider what gifts and knowledge were brought forward within each research team and advisory circles, and whether the perspectives and gifts of others were missing from such groups. In the light of existing Western and Indigenous GBA frameworks, we note that the concepts of intersectionality and distinction-based approaches are used as heuristics to help their users consider the differential *impacts* that legislation, policy, or initiatives may have on people’s lives, according to their intersecting identity factors, life experiences, and contexts [22,38]. Our findings add to the existing principles of Indigenous and Western GBA frameworks by highlighting the importance of considering the intersecting identity factors among those who *design and advise* the ongoing development of research processes. In bringing together the gifts of both Indigenous and non-Indigenous perspectives within research, we are reminded that community contexts and preferences should be the drivers of research design within Indigenous GBA [22], that Indigenous-specific methods of data collection and analysis do not need Western methods to be validated [68], and that an honorable interaction between Indigenous and Western knowledge systems must take place within an ‘ethical space’ that prevents the appropriation of Indigenous knowledge, the imposition of Western values, and that supports the development of Indigenous-specific ways of knowing and living [69].

Secondly, learning about Indigenous GBA may be as much about trusting the embodied knowledge and lived experience of all the *A SHARED Future* members as it is about navigating the abundant formal literature on Indigenous GBA. Research findings showed that the initial efforts of the Gender Champions in synthesizing existing knowledge on Indigenous GBA within the Living Compendium did not have the intended effect on building the knowledge and capacity of each research team on Indigenous GBA. Looking back, we note that the Living Compendium as a knowledge translation tactic was similar to mainstream GBA learning products that provide a series of definitions of GBA-related concepts, and a series of questions or tasks to consider for individual researchers to apply into their work [70,71]. Our research findings indicate that more relational ways of knowledge sharing—such as hosting conversations about each other’s experience with Indigenous GBA and providing supportive feedback on each research project—were more successful means for supporting the *A SHARED Future* research teams to center culturally-grounded gender considerations in their research projects. Becoming acquainted with the literature on Indigenous GBA can be extremely helpful for identifying the key principles and practices for centering culturally grounded considerations within research. However, the principle of *Etuaptmumk* can guide research teams to recognize the gifts that each person carries, such as their knowledge, lived experience, and community relations, as the primary basis for operationalizing Indigenous GBA within the specific context of their research projects.

Drawing on the previous two lessons, the authors’ third takeaway is that Indigenous GBA *implementation* must transcend the procedural, overcoming the methodological notion of gender-based *analysis* and toward its replacement with an ethical orientation to gender-based *relationality*. Gender-based relationality goes beyond the concept of ‘gender relations’ by asserting that research methods and their intended outcomes should contribute to and foster more respectful, balanced, and accountable relationships across genders, and relationships to the land. Indigenous GBA should not be seen as a ‘component’ of the research process but a fundamental way of doing research with others. Existing frameworks such as the Métis-Specific GBA+ Tool posit that the implementation of Indigenous GBA requires a significant amount of time dedicated to building and sustaining relationships with community members and leaders, as well as ongoing consultation and direction [21]. Similarly, relationships at the community level have allowed *A SHARED Future* researchers to identify key community members who could provide guidance, and the appropriate protocol to respectfully engage people and the land through research relationships. However, even with strong community advice, the research preparation activities and fieldwork alike presented researchers with significant challenges when discussing issues of white privilege and male privilege, as heteropatriarchy and settler colonialism are intimately intertwined in framing the lives of Indigenous and settler peoples alike [72]. Here, the *Etuaptmumk* principle reminds us that bringing the strengths of multiple perspectives is not a smooth one-time procedure, but rather a long-term journey of collective learning [27].

In discussing the three main lessons from our experience in the implementation of Indigenous GBA in research, we recognize the limitations of this paper. The number of data gathering activities has been limited until this point. *A SHARED Future* activities are ongoing, and, with the emergence of the COVID-19 pandemic, the hosting of follow-up interviews and additional sharing circles has been cancelled. In many cases, the Indigenous *A SHARED Future* researchers and community co-leads have contributed to the COVID-19 responses of their communities. Considering the circumstances of constrained capacity, collaborative writing has strengthened the empirical grounding of this paper, as each co-author’s input was based on their research experience within *A SHARED Future*. We believe that the novel nature of the insights shared by each other through a reflexive writing process merits its detailed discussion within this paper. Future steps for our team include creating thick descriptions about the operationalization of Indigenous GBA within the specific First Nation, Inuit, or urban Indigenous community contexts of our constituent research teams. While this paper has focused on the views of the PIs, the authors hope that future work highlights the perspectives of the Indigenous co-leaders of each program, research trainees, non-Indigenous collaborators, and local Elders or knowledge holders. Future inquiry on the learning and implementation of Indigenous GBA within research will provide novice and seasoned researchers alike with invaluable support to redress the damaging legacies of a patriarchal and colonial gaze within research.

## 5. Conclusions

This paper addresses a gap in practical guidance for research teams to embrace Indigenous GBA principles according to the unique histories, contexts, participants, and ways of knowing involved in each research project. In the case of *A SHARED Future* as a multi-site research program led by Indigenous and non-Indigenous researchers, the Mi’kmaw principle of *Etuaptmumk* guided the authors’ journey in taking Indigenous GBA principles into practice. From our research findings, we highlighted three central lessons from our learning journey into Indigenous GBA implementation: (1) to consider the roles of intersecting identity factors of those who design and advise the research process in the framing of all aspects of the research design; (2) to decenter the role of written ‘knowledge products’ in building researchers’ capacity to understand and undertake Indigenous GBA, in favour of relational ways of knowledge sharing; and (3) to realize that data analysis is only a small part of Indigenous GBA implementation, and that a commitment toward embodying and promoting respectful gender-based relationality in all research processes and outcomes is at the forefront of Indigenous GBA implementation. The authors hope that the successes and failures of our research program support other researchers, practitioners, and communities in their journeys of integrating the principles of Indigenous GBA frameworks to their research projects. Supporting the development and uptake of Indigenous GBA frameworks at an international level can help to interrupt the imposition of colonial and patriarchal worldviews through research and support Indigenous resurgence processes throughout the globe.

## Data Availability

The data presented in this study is not publicly available for ethical reasons. Participants were assured that research data would only be available to members of the research team and destroyed upon five years of project completion.

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
