# Peer review of "Implementing Indigenous Gender-Based Analysis in Research: Principles, Practices and Lessons Learned"

_ijerph, 2021, doi:10.3390/ijerph182111572_

Round 1

Reviewer 1 Report

First of all, I thank the authors of this work. It seems original to me in terms of the methodological proposal and I have enjoyed reading it. I admit that the research process has some limitations that are recognized by the authors themselves. Therefore, I do not consider them as limitations; but as contributions, because they will contribute to improving the process in the future. Good luck

Author Response

Dear reviewer,

Thank you for reviewing our manuscript. We appreciate your kind words about our work.

The authors,

Reviewer 2 Report

This paper discusses the principles, practices, and lessons learned from the implementation of Indigenous gender-based analysis in research.  Authors assert a need for this study by stating that while numerous tools for assessing gender inequality have emerged across the globe, many have failed to consider the impacts of colonialism on Indigenous people’s lives and lands.  Authors aim to address a gap in the literature by proving guidance on how to incorporate Indigenous gender-based frameworks in the context of research.

The authors take on a laudable effort to address the exclusion of Indigenous perspectives in gender-based policymaking, which they describe as being a notable deficiency across the globe.  The following feedback is offered:

  1. Authors present a compelling need for this research in the Introduction. What may also be helpful is to provide a brief background and overview of Indigenous peoples in Canada.  What is their history?  What are their demographics?  What part of the country do they commonly inhabit?  Who would be considered an Indigenous person? This may be helpful information for the lay reader.
  2. Line 58 – What is meant by being ‘Two-Spirit’? A brief definition may be helpful.
  3. Lines 252-253 – The sentence “In early 2019, the Gender Champions conducted semi-structured video call interviews with four A SHARED Future’s PIs; five PIs were involved at this stage” is a bit unclear. There were interviews conducted with 4 PIs yet 5 PIs were involved?
  4. Concerning the methods, it would be helpful to provide more insight into the participant recruitment. How were the PIs recruited and over what time period?
  5. The quotes presented in the Findings section are meaningful. Perhaps they can be presented in bulleted or table format to facilitate readability.
  6. Overall, this is an insightful, pertinent, and unique study. Attending to some clarifying questions can help to improve the quality of the paper.

Author Response

Dear Reviewer,

Thank you for your feedback. We have incorporated your suggestions into our manuscript. Please see the attachment.

Kind regards,

Reviewer 3 Report

I fully agree with the authors that the exclusion of Indigenous perspectives in gender-based policymaking is massive debt of policy makers, not only in Canada, but also in New Zealand, or countries in South America such as Chile and Brazil, which are also former colonies. It would be good to stress this point: this is not only a problem in Canada.

My only important suggestion, therefore, would be to make some general references to this lacuna for other countries beyond Canada, that would make the argument and the paper stronger. For example, in the conclusions, the practical guidance for research teams should be addressed to teams beyond Canada.

Linked to this, despite these shortcomings in Canada, it would be would to know a bit more about the situation in other countries, in particular New Zealand, which enjoy a very advanced agenda on Indigenous people's policies. This, at a very general level, only based on secondary literature.

Something minor. Could you please clarify how the Interviewee of the semi-structured interviews with four A SHARED Future’s PIs were selected? Wy those 4? Which was the universe? Thanks

Also, I think the "Collaborative, reflexive writing" was a very good idea. Could you tells us a bit more? Is there any similar experience elsewhere? In particular for published academic papers, thanks

Other than that, the methodology is very sounds, so many congrats to the research team.

Author Response

(The authors gave the same response as above.)
